Utilizing network pharmacology and molecular docking to explore the underlying mechanism of Guizhi Fuling Wan in treating endometriosis

Wang Haoxian 1
Zhou Gang 2
Zhuang Mingyan 3
Wang Wei doctor.vi@163.com 2
Fu Xianyun dinnar1@163.com 1
1 Medical College, China Three Gorges University , Yichang , China
2 College of Traditional Chinese Medicine, Three Gorges University & Yichang Hospital of Traditional Chinese Medicine , Yichang , China
3 Maternity and Child Health Care Hospital, Three Gorges University , Yichang , China
Wang Yong
Electronic publication date: 2021 Apr 2
Publication date: 2021
Volume: 9
Electronic Location ID: e11087
Received 2020 Jun 9; Accepted 2021 Feb 19
Copyright: ©2021 Wang et al.
Copyright year: 2021
Copyright holder: Wang et al.
License: This is an open access article distributed under the terms of the Creative Commons Attribution License, which permits unrestricted use, distribution, reproduction and adaptation in any medium and for any purpose provided that it is properly attributed. For attribution, the original author(s), title, publication source (PeerJ) and either DOI or URL of the article must be cited.
License URL: https://creativecommons.org/licenses/by/4.0/

Keywords: Herbal medicine, Complementary medicine, Natural product, Endometriosis, Network pharmacology, Molecular docking

Funding: The National Natural Science Foundation of China No. 81973897 Scientific Research Foundation of HuBei provincial health department No. WJ2019F079 Yichang Medical and Health Research Foundation No. A19-301-44 The study was supported by the National Natural Science Foundation of China (No. 81973897), the Scientific Research Foundation of HuBei provincial health department (No. WJ2019F079), and the Yichang Medical and Health Research Foundation (No. A19-301-44). The funders had no role in study design, data collection and analysis, decision to publish, or preparation of the manuscript.

==============================
Background

Guizhi Fuling Wan (GZFLW) is a widely used classical Chinese herbal formulae prescribed for the treatment of endometriosis (EMs). This study aimed to predict the key targets and mechanisms of GZFLW in the treatment of EMs by network pharmacology and molecular docking.

Methods

Firstly, related compounds and targets of GZFLW were identified through the TCMSP, BATMAN-TCM and CASC database. Then, the EMs target database was built by GeneCards. The overlapping targets between GZFLW and EMs were screened out, and then data of the PPI network was obtained by the STRING Database to analyze the interrelationship of these targets. Furthermore, a topological analysis was performed to screen the hub targets. After that, molecular docking technology was used to confirm the binding degree of the main active compounds and hub targets. Finally, the DAVID database and Metascape database were used for GO and KEGG enrichment analysis.

Results

A total of 89 GZFLW compounds and 284 targets were collected. One hundred one matching targets were picked out as the correlative targets of GZFLW in treating EMs. Among these, 25 significant hub targets were recognized by the PPI network. Coincidently, molecular docking simulation indicated that the hub targets had a good bonding activity with most active compounds (69.71%). Furthermore, 116 items, including the inflammatory reaction, RNA polymerase, DNA transcription, growth factor activity, and steroid-binding, were selected by GO enrichment analysis. Moreover, the KEGG enrichment analysis results included 100 pathways focused on the AGE-RAGE pathway, HIF pathway, PI3K Akt pathway, MAPK pathway, and TP53 pathway, which exposed the potential mechanisms of GZFLW in treating EMs. Also, the MTT colorimetric assay indicated that the cell proliferation could be inhibited by GZFLW. Compared with the control group, the protein levels of P53, BAX, and caspase3 in the drug groups were all increased in Western blotting results. The results of flow cytometry showed that the percentage of apoptotic cells in the GZFLW group was significantly higher than that in the control group.

Conclusion

Through the exploration of network pharmacology and molecular docking technology, GZFLW has a therapeutic effect on EMs through multi-target mechanism. This study provided a good foundation for further experimental research.

Introduction

Endometriosis (EMs), a common painful gynecological disease, occurs in 10% women of childbearing age (Reid et al., 2019). The most frequent symptoms of EMs include dysmenorrhea, chronic pelvic pain, and infertility (Bruun et al., 2018; Lalani et al., 2018). The incidence of developing many comorbidities, including irritable bowel syndrome, constipation, ovarian cancer, endometrial cancer, is significantly higher among EMs patients comparing to the general population (Schomacker et al., 2018). Although current therapies, including surgery, non-steroidal anti-inflammatory drugs (NSAIDs), hormone treatments, and so on, could relief some significant symptoms of EMs, these treatments have also led to some side effects and a high recurrence rate (Prefumo & Rossi, 2018; Rabinerson, Hiersch & Gabbay-Ben-Ziv, 2018). The patient’s quality of life declines, and half of them are not satisfied with the available medical support (Verket et al., 2018). Therefore, it is still necessary to find novel and useful treatment methods (Lukas et al., 2018).

Historical Chinese medical texts have documented the use of the traditional Chinese herbal formula Guizhi Fuling Wan (GZFLW) for the EMs-like symptoms such as dysmenorrhea since the late Eastern Han Dynasty (200-210AD), and it is still widely used today for the treatment of EMs (Li et al., 2018; Wu et al., 2015; Zhao, 2016) due to excellent therapeutic effect, low side effects (Wang et al., 2018), and safety. GZFLW consists of the original powder of five natural plants, including Cinnamon Twig (CT), Poria Cocos (PC), Cortex Moutan (CM), Radix Paeoniae Rubra (RPR), and Peach Kernel (PK). Animal experiments have confirmed that GZFLW could relieve dysmenorrhea effectively (Lang et al., 2018; Yang, 2019). Nevertheless, the complex pathogenesis of EMs and the multiplex mechanism of GZFLW remains unclear. The principles and mechanisms by which GZFLW is useful for treating EMs need to be uncovered.

The holistic treatment of traditional Chinese medicine (TCM) has attracted more and more attention. The mechanism of drug therapy has shifted from single target to multiple interacting targets mediated by multiple compounds. Nowadays, the extensive application of network pharmacology and molecular docking provides a more effective method for the research and evaluation of the multi-target effect of multi-component drugs on diseases, which can reveal the mechanism of TCM treatment from a holistic perspective (Hopkins, 2008). In the progress of bioinformatics and pharmacy, network pharmacology has become a capable vehicle to reveal the compatibility mechanism of TCM prescription (Fang et al., 2017; Ming et al., 2017; Zhao et al., 2015). An increasing number of researches have used them to analyze the possible molecular mechanisms of TCMs. This systematic research conception is consistent with the holistic theory and the synergistic mechanism of TCM (Lang et al., 2018). However, the possible mechanism of GZFLW treating EMs has not been systematically studied by network pharmacology.

Therefore, in this study, we screened multiple databases to find the active components of GZFLW and its possible targets for treating EMs. Target genes of GZFLW and EMs were matched to obtain overlapping results. PPI network data was obtained through the STRING database, and the hub targets were screened through topology analysis. After that, molecular docking technology was used to screen and verify the binding degree of the main active compounds and hub targets. Finally, the DAVID database and Metascape database were used for gene ontology (GO) and Kyoto Encyclopaedia of Genes and Genomes (KEGG) enrichment analysis. GO is generally used to describe the genes function and genes relationships (Ashburner et al., 2000), and KEGG is used for the enrichment of functions and signaling pathways (Kanehis & Goto, 2000). The vital active ingredients and mechanism of GZFLW in the treatment of EMs were confirmed by molecular docking technology. The workflow is shown in Fig. 1.

Figure 1 The whole framework of the research process.

Materials & Methods

Chemical ingredients database building

To collect the chemical ingredients of the five herbs contained in GZFLW, the Traditional Chinese Medicine Systems Pharmacology Database (Ru et al., 2014) (TCMSP, https://tcmspw.com/tcmsp.php), the Bioinformatics Analysis Tool for Molecular Mechanism of Traditional Chinese Medicine (Liu et al., 2016) (BATMAN-TCM, http://bionet.ncpsb.org/batman-tcm/index.php) and Chinese Academy of Sciences Chemistry Database (Wang et al., 2015) (CASC, http://www.organchem.csdb.cn/scdb/default.htm) were used, which are both bioinformatics analysis tools for the main components of TCM. Four hundred ninety-four herbal ingredients were screened out.

Active ingredients screening

According to the characteristics of absorption, distribution, metabolism, and excretion (ADME) of drug, oral bioavailability (OB), and drug-likeness (DL) were used as screening indexes. OB is the percentage of oral drugs absorbed into the bloodstream, which is a frequently-used pharmacokinetic parameter. The OB was obtained using the OBioavail1.1 software, which covers 805 different drug and drug-like molecules (Liu et al., 2013). OB is a crucial indicator to judge whether the active ingredient can become a feasible therapeutic molecule (Xu et al., 2012). DL is a parameter to measure the ADME of drug molecules, which could help optimize pharmacokinetic and pharmaceutical properties. The DL threshold is 0.18 (Liu et al., 2013), which depends on the average of Drugbank (https://www.drugbank.ca/). DL is often used to select the active ingredients with “drug-like” properties in TCM composition (Tao et al., 2013). The OB and DL indices of all the related ingredients are presented in the TCMSP. In this process, those ingredients with OB ≧30% and DL ≧0.18 were chosen as the potential effect components for the next step.

GZFLW targets prediction

The active compounds’ targets in GZFLW were obtained from TCMSP, BATMAN-TCM, and CASC database, with the species limited as “Homo sapiens”.

Endometriosis therapeutic targets database building

Related targets of EMs were screened out by the GeneCards Database (Stelzer et al., 2016) (https://www.genecards.org/). The GeneCards database is often used to predict genetic information related to human diseases. We searched the database with the keyword “endometriosis” to obtain targets. Finally, the targets of GZFLW active ingredients were matched with the therapeutic targets of EMs. The overlapping targets were chosen as the potential targets of GZFLW in the treatment of EMs. These targets were then uploaded to STRING Database (Szklarczyk et al., 2017) (http://string-db.org/) to obtain the interactions of the screened targets with the confidence ≧0.7 and the result of protein-protein interaction (PPI).

Network topological feature set definitions

We chose two parameters to evaluate the topological characteristics: “Degree” reflects the number of other nodes interacting with this node; “Betweenness Centrality” (BC) is measured by the percentage of all shortest paths. The nodes with high betweenness can significantly impact the net by controlling the information transmitted between other nodes. The node parameters are positively correlated with their topology importance in the network.

Molecular docking of hub targets and active ingredients

Molecular docking is a computational tool to predict the binding ability and connection type of proteins and ligands. It can calculate and predict the conformation and direction of ligands at active protein sites. AutoDock 1.5.6 (http://autodock.scripps.edu/) was a molecular docking software that could be used to dock the hub targets and active ingredients based on network pharmacology. In this docking process, the 3D structure of thirteen hub targets were retrieved from RCSB Protein Data Bank (PDB) (http://www.rcsb.org/): AKT1 (PDB ID: 4EKL, 6S9W), TNF (PDB ID: 2E7A, 2ZJC, 2AZ5), TP53 (PDB ID: 4MZI, 6FF9), VEGFA (PDB ID: 4WPB, 3QTK, 4QAF), MAPK1 (PDB ID: 4G6N, 1WZY), MMP9 (PDB ID: 6ESM, 4WZV), JUN (PDB ID: 5T01, 1T2K), MAPK8 (PDB ID: 3VUK, 4L7F), INS (PDB ID: 4AJX, 4CY7), EGF (PDB ID: 1JL9,), IL6 (PDB ID: 4O9H, 1ALU), PTGS2 (PDB ID: 5F19), and FOS (PDB ID: 1FOS); the 3D shapes of active compounds were provided from ZINC Database (http://zinc.docking.org/) and PubChem Database (https://pubchem.ncbi.nlm.nih.gov/). Binding energy was used as docking score to evaluate the protein-ligand binding potential of molecular docking. Among them, those results with value ≤ -5 were selected and considered to have moderate binding potential and tight combination.

Enrichment analysis

We used the DAVID (https://david.ncifcrf.gov/) (Huang, Sherman & Lempicki, 2008) for GO enrichment analysis, including biological process (BP), molecular function (MF) and cellular component (CC). KEGG enrichment analysis was performed using the Metascape Database to obtain potential target-pathways (Zhou et al., 2019) (http://metascape.org/).

Network construction

Networks, including the hub target-compound and target-pathway, were then constructed. All networks were built using Cytoscape 3.7.2 (http://www.cytoscape.org/) (Shannon et al., 2003), an open-source software platform for visualization and data analysis of complex networks.

Primary endometrial stromal cells culture

HEM15a cells were cultured in DMEM containing 10% FBS and 1% penicillin/streptomycin at 37 ∘C in the cell incubator with a humid atmosphere containing 5% CO 2. After that, the vimentin staining of cells was identified by immunofluorescence.

MTT colorimetric assay

The hEM15a cells were seeded in a 96-well plate and divided into the control group and the drug group. After incubation at 37 ∘C for 48 h and reacted with GZFLW of different concentrations, hEM15a cells were reacted with 10 μL MTT solution. Finally, 150 μL dimethylsulfoxide (DMSO) was added after reacting for 4 h. The optical absorbance was detected at 568 nm by a plate reader.

Western blotting

The hEM15a cells were splitted in RIPA buffer (Beyotime) added with 10 μl PMSF (Aladdin) and phosphatase inhibitors (Beyotime) to extract total proteins. After that, equivalent amounts of proteins were resolved by poly acrylamide gel electrophoresis (PAGE) and transferred to polyvinylidene fluoride (PVDF) membranes (Millipore, Massachusetts, USA). Then, TBST with 5% skim milk was adopted for blocking. Afterward, the membranes were reacted with primary antibodies against GAPDH (1:1000, AB-P-R 001, Xianzhi, Hangzhou, China), BAX (1:1000, Ab32503, Abcam, Britain), P53 (1:1000, 10442-1-AP, Sanying, Wuhan, China), and caspase3 (1:2000, Ab184787, Abcam), at 4 ∘C overnight. Subsequently, the matched secondary antibody was added to the membranes.

Apoptosis analysis by flow cytometry

The apoptosis of hEM15a cells was evaluated by Annexin V-FITC/PI apoptosis kit (KaiJi, Nanjing, China). The collected hEM15a cells, divided into normal group and GZFLW group, were resuspended in 500 ul binding buffer and added with Annexin V-PI solution. Then, the hEM15a cells were incubated for 5∼15 min without light. The percentage of apoptotic cells was then immediately detected on a flow cytometry (Beckmancoulter).

Results

Active compounds of GZFLW

By retrieving from the TCMSP, BATMAN-TCM, and CASC database, there were 494 related ingredients of GZFLW in total, and there were 220 (44.5%) of CT, 34 (6.9%) of PC, 55 (11.1%) of CM, 119 (24.1%) of RPR, 66 (13.3%) of PK. With OB ≧30% and DL ≧ 0.18 as indexes, 89 active ingredients were screened out (Table 1), and the Herbs-Compounds network was constructed as Fig. 2.

Target prediction analysis

In this process, we collected 284 targets of 89 active ingredients, and there were 58 in CT, 48 in PC, 185 in CM, 104 in RPR, and 62 in PK. Via the keyword of “endometriosis,” the therapeutic targets of EMs were obtained from GeneCard Database with a total of 1350. One hundred one overlapping targets were then obtained as the related targets of GZFLW in the treatment of EMs (Fig. 3).

The data of the PPI network of those 101 targets were subsequently obtained in STRING Database. With confidence ≧0.7, there were 95 nodes and 751 edges in total. Taking two essential parameters of “degree” and “betweenness” as screening indexes, the topological analysis of targets mentioned above was performed. Targets more significant than or equal to the median are used as hub targets for GZFLW against EMs. The screened thresholds were degree ≧11.5 and betweenness ≧0.007, and the results were 25 hub nodes with 234 edges, including IL6, JUN, TNF, MAPK1, TP53, EGF, MAPK8, MMP9, VEGFA, AKT1, INS, FOS, ICAM1, PTGS2, CCL2, CCND1, EGFR, IL1B, MYC, IL10, PTEN, ESR1, PPARG, RELA and MMP2 (Table 2). When the 25 significant hub nodes and other 70 nodes were distributed with “degree” and “betweenness”, the network of 95 nodes was built as Fig. 4.

Based on the 25 key targets and related 29 active ingredients, we further established the network of Hub nodes—Compounds (Fig. 5). Among them, quercetin (MOL000098) is associated with 21 hub targets, and kaempferol (MOL000422) interrelates to 7 key targets. Besides, baicalein (MOL002714) is related to 6 key targets, while pachymic acid (MOL000289), ellagic acid (MOL001002), and taxifolin (MOL004576) act on 3 key targets respectively.

Molecular Docking

Thirteen hub targets with top degrees of GZFLW were identified with seven active compounds by AutoDock. There were 122 results (69.71%) of them had a moderate binding potential, which indicated that active ingredients of GZFLW could well bind to the targets for the treatment of EMs (Table 3). The ligands are mainly linked with corresponding proteins and critical amino acids around them in the form of hydrogen bonds (Fig. 6).

GO and KEGG pathway enrichment analysis

With the database of DAVID and Metascape, the enrichment analysis on 101 targets was performed and resulted in 116 GO items and 100 KEGG pathways.

GO enrichment analysis

After 116 items were sorted in descending order based on P-value, the first eight items of three parts, BP, MF, and CC, were selected (Fig. 7). In the aspect of BP, we mainly had: inflammatory response (GO:0006954), positive regulation of transcription from RNA polymerase II promoter (GO:0045944), transcription DNA-templated (GO:0006351), lipopolysaccharide-mediated signaling pathway (GO:0031663), positive regulation of cell division (GO:0051781), negative regulation of growth of symbiont in host (GO:0044130), response to toxic substance (GO:0009636) and positive regulation of cytokine secretion (GO:0050715); in the part of MF, we obtained heme binding (GO:0020037), sequence-specific DNA binding (GO:0043565), transcription factor activity sequence-specific DNA binding (GO:0003700), growth factor activity (GO:0008083), identical protein binding (GO:0042802), protein homodimerization activity (GO:0042803), cytokine activity (GO:0005125) and steroid binding (GO:0005496); in the aspect of CC, there were extracellular space (GO:0005615), nucleus (GO:00056340), cytosol (GO:0005829), membrane raft (GO:0045121), apical plasma (GO:0016324), external side of plasma membrane (GO:0009897), endoplasmic reticulum membrane (GO:0005789) and extrinsic component of external side of plasma membrane (GO:0031232). Based on the above three aspects, it is possible that the mechanism of GZFLW in treating EMs was the result of multi-pathway synergy.

Table 1 Information for 89 active ingredients.

Herb name	Mol ID	compound	Code name	OB/%	DL/%	
Cinnamon Twig	MOL001736	(-)-taxifolin	CT-1	60.51	0.27	
	MOL004576	taxifolin	CT-2	57.84	0.27	
	MOL000492	(+)-catechin	CT-3	54.83	0.24	
	MOL000073	ent-Epicatechin	CT-4	48.96	0.24	
	MOL000358	beta-sitosterol	CT-5	36.91	0.75	
	MOL000359	sitosterol	CT-6	36.91	0.75	
	MOL000991	cinnamaldehyde	CT-7	31.99	0.02	
Paria cocos	MOL000282	ergosta-7,22E-dien-3beta-ol	PC-1	43.51	0.72	
	MOL000283	Ergosterol peroxide	PC-2	40.36	0.81	
	MOL000275	trametenolic acid	PC-3	38.71	0.8	
	MOL000296	hederagenin	PC-4	36.91	0.75	
	MOL000289	Pachymic Acid	PC-5	33.63	0.81	
	MOL000273	(2R)-2-[(3S,5R,10S,13R,14R,16R,17R)-3, 16-dihydroxy-4,4,10,13,14-pentamethyl-2,3, 5,6,12,15,16,17-octahydro-1H-cyclopenta[a]phenanthren-17-yl] -6-methylhept-5-enoic acid	PC-6	30.93	0.81	
Cortex Moutan	MOL000211	Mairin	CM-1	55.38	0.78	
	MOL000492	(+)-catechin	CM-2	54.83	0.24	
	MOL000098	quercetin	CM-3	46.43	0.28	
	MOL000422	kaempferol	CM-4	41.88	0.24	
	MOL000359	sitosterol	CM-5	36.91	0.75	
	MOL000874	Paeonol	CM-6	28.79	0.04	
Radix Paeoniae Rubra	MOL001918	paeoniflorgenone	RPR-1	87.59	0.37	
	MOL006992	(2R,3R)-4-methoxyl-distylin	RPR-2	59.98	0.3	
	MOL000492	(+)-catechin	RPR-3	54.83	0.24	
	MOL001924	paeoniflorin	RPR-4	53.87	0.79	
	MOL000449	Stigmasterol	RPR-5	43.83	0.76	
	MOL001002	ellagic acid	RPR-6	43.06	0.43	
	MOL004355	Spinasterol	RPR-7	42.98	0.76	
	MOL002776	Baicalin	RPR-8	40.12	0.75	
	MOL005043	campest-5-en-3beta-ol	RPR-9	37.58	0.71	
	MOL006999	stigmast-7-en-3-ol	RPR-10	37.42	0.75	
	MOL000358	beta-sitosterol	RPR-11	36.91	0.75	
	MOL000359	sitosterol	RPR-12	36.91	0.75	
	MOL002714	baicalein	RPR-13	33.52	0.21	
	MOL002883	Ethyl oleate (NF)	RPR-14	32.4	0.19	
Peach kenel	MOL001351	Gibberellin A44	PK-1	101.61	0.54	
	MOL001353	GA60	PK-2	93.17	0.53	
	MOL001349	4a-formyl-7alpha-hydroxy-1-methyl-8-methylidene-4aalpha, 4bbeta-gibbane-1alpha,10beta-dicarboxylic acid	PK-3	88.6	0.46	
	MOL001344	GA122-isolactone	PK-4	88.11	0.54	
	MOL001329	2,3-didehydro GA77	PK-5	88.08	0.53	
	MOL001360	GA77	PK-6	87.89	0.53	
	MOL001340	GA120	PK-7	84.85	0.45	
	MOL001339	GA119	PK-8	76.36	0.49	
	MOL001358	gibberellin 7	PK-9	73.8	0.5	
	MOL001342	GA121-isolactone	PK-10	72.7	0.54	
	MOL001361	GA87	PK-11	68.85	0.57	
	MOL001355	GA63	PK-12	65.54	0.54	
	MOL001352	GA54	PK-13	64.21	0.53	
	MOL001328	2,3-didehydro GA70	PK-14	63.29	0.5	
	MOL001323	Sitosterol alpha1	PK-15	43.28	0.78	
	MOL001368	3-O-p-coumaroylquinic acid	PK-16	37.63	0.29	
	MOL000493	campesterol	PK-17	37.58	0.71	
	MOL000296	hederagenin	PK-18	36.91	0.75	
	MOL000358	beta-sitosterol	PK-19	36.91	0.75	
	MOL001320	Amygdalin	PK-20	4.42	0.61	

KEGG pathway enrichment analysis

To further illustrate the potential mechanism of GZFLW in the treatment of EMs, we performed the KEGG pathway enrichment analysis on 101 targets. We selected the top 20 pathways based on the P-value, such as the AGE-RAGE signaling pathway(hsa04933), HIF-1 signaling pathway (hsa04066), PI3K-Akt signaling pathway (hsa04151), MAPK signaling pathway (hsa04010), EGFR tyrosine kinase inhibitor resistance (hsa01521) (Fig. 8). Then, we constructed the Target-Pathway Network to intuitively reveal the relationship between the hub targets and pathways (Fig. 9).

Figure 2 Herbs-Compounds Network.

The yellow nodes represent herbs in GZFLW, and the blue nodes represent active compounds. CT represents Cinnamon Twig, PC represents Paria cocos, CM represents Cortex Moutan, RPR represents Radix Paeoniae Rubra, and PK represents Peach kenel. The edges represent the relationship between them.

Primary endometrial stromal cells culture

The primary endometrial stromal cells were spindle-shaped or star-shaped with large and round nuclei, and proliferative fibrous tissue could be seen. After that, the immunofluorescence was used to identified the vimentin staining of cells in different groups (Fig. 10).

Figure 3 The Venn diagram of the targets both in endometriosis targets and GZFLW targets.

MTT colorimetric assay

In comparison with the control group, in addition to 0.5 mg/ml, the other different doses of treatment groups (1 mg/ml, 3 mg/ml, 5 mg/ml, 10 mg/ml, 20 mg/ml) had significant inhibitory effect on the proliferation of endometrial stromal cells (P < 0.05, 0.01 or 0.001; Fig. 11).

Table 2 Information for 25 hub targets.

NO.	Gene	Degree	BC	NO.	Gene	Degree	BC	NO.	Gene	Degree	BC	
1	IL6	23	9.1459	10	AKT1	20	6.5181	19	MYC	17	3.4007	
2	JUN	23	8.9097	11	INS	20	5.9093	20	IL10	16	2.4666	
3	TNF	23	8.8208	12	FOS	19	6.1117	21	PTEN	16	3.0134	
4	MAPK1	22	8.6040	13	ICAM1	18	3.4111	22	ESR1	15	1.5358	
5	TP53	22	7.9533	14	PTGS2	18	5.1667	23	PPARG	15	2.2070	
6	EGF	21	6.3421	15	CCL2	17	2.6724	24	RELA	15	2.6052	
7	MAPK8	21	7.5937	16	CCND1	17	3.3755	25	MMP2	14	2.0489	
8	MMP9	21	6.9767	17	EGFR	17	3.4135					
9	VEGFA	21	8.0120	18	IL1B	17	3.2828					

Figure 4 The network of 95 nodes.

The different color represents different nodes from each ingredients: The blue nodes represent the targets from CM, the orange nodes represent the targets from PC, the purple nodes represent the targets from RPR, and the yellow nodes represent the targets that are targeted by more than one ingredient. The node size is proportional to the target degree in the network. The edge color changes from light to dark reflect the betweenness value changes from low to high in the network.

Figure 5 Hub nodes-Compunds Network.

The circle nodes represent the significant hub nodes, and the diamond nodes represent the compounds. The node size is proportional to the target degree in the network.

Table 3 The docking information of 13 targets with the corresponding compounds.

The values in this table are docking scores that between targets ans compounds.

	Pachymic Acid	taxifolin	quercetin	paeoniflorin	beta-sitosterol	campesterol	hederagenin	
AKT1-4EKL	−5.02	−7.65	−5.58	−4.19	−8.56	−8.27	−7.61	
AKT1-6S9W	−7.85	−6.88	−5.08	−4.21	−8.59	−9.99	−8.79	
MAPK1-6RQ4	−7.25	−6.22	−5.26	−3.61	−6.25	−7.16	−8.16	
MAPK1-1WZY	−6.62	−6.51	−4.3	−3.7	−8.09	−6.64	−8.02	
TNF-2E7A	−8.68	−6.61	−7.22	−4.2	−9.17	−10.31	−10.69	
TNF-2ZJC	−4.25	−5.06	−3.45	−2.55	−4.9	−5.44	−4.98	
TNF-2AZ5	−6.07	−4.57	−4	−3.55	−6.08	−7.16	−6.79	
TP53-4MZI	−7.28	−7.11	−5.71	−5.12	−7.65	−6.27	−7.84	
TP53-6FF9	−5.44	−4.19	−4.07	−2.93	−5.62	−5.14	−6.28	
VEGFA-4WPB	−6.7	−5.81	−5.36	−4.09	−7.48	−7.01	−7	
VEGFA-3QTK	−4.31	−3.99	−3.15	−2.58	−4.47	−6.18	−5.61	
VEGFA-4QAF	−4.76	−5.41	−3.96	−2.4	−5.93	−5.64	−5.58	
MMP9-6ESM	−7.83	−9.32	−7.55	−6.24	−9.03	−9.36	−8.83	
MMP9-4WZV	−6.09	−7.53	−7.53	−4.45	−10.01	−10.5	−6.84	
JUN-5T01	−5.46	−4.15	−4.63	−1.96	−5.51	−4.85	−7.6	
JUN-1T2K	−6.28	−3.07	−3.3	−1.81	−5.17	−5.03	−6.43	
MAPK8-3VUK	−6.25	−5.87	−6.17	−4.31	−7.05	−8.9	−6.99	
MAPK8-4L7F	−4.96	−5.94	−4	−3.12	−5.5	−5.34	−6.96	
INS-4AJX	−8.6	−6.48	−6.66	−6.32	−8.72	−9.18	−7.92	
INS-4CY7	−6.62	−4.84	−5.6	−4.13	−6.56	−6.08	−7.58	
EGF-IJL9	−7.42	−7.12	−7.29	−3.89	−7.94	−7.58	−9.14	
IL6-4O9H	−6.49	−5.34	−4.66	−3.81	−5.96	−5.95	−6.48	
IL6-1ALU	−5.23	−3.94	−3.74	−2.5	−4.58	−5.09	−6.45	
PTGS2	−7.44	−5.77	−4.24	−2.86	−5.85	−5.87	−9.63	
FOS	−5.71	−3.17	−3.57	−1.46	−3.63	−6.04	−6.1	

Figure 6 The molecular docking.

(A) TNF (PDBID: 2E7A): campesterol molecular docking. (B) TNF (PDBID: 2E7A): beta-sitosterol molecular docking. (C) MMP9 (PDBID: 6ESM): campesterol molecular docking. (D) MMP9 (PDBID: 6ESM): beta-sitosterol molecular docking. The yellow sticks represent TNF and VEGFA. The blue sticks represent active compounds.

Figure 7 The GO enrichment analysis of 101 nodes.

The orange part represents the Biological process. The blue part represents the Cellular component. The green part represents the Molecular function.

Figure 8 The top 20 pathways of KEGG enrichment.

Figure 9 Target-Pathway Network.

The orange circle nodes represent the hub nodes, and the pink nodes represent the other nodes. The red diamond nodes represent the related pathways. The node size is proportional to the target degree in the network.

Figure 10 Primary endometrial stromal cells and vimentin staining.

(A) Primary endometrial stromal cells. (B) The vimentin staining of cells.

Figure 11 The results of MTT colorimetric assay.

The cell proliferation of hEM15a cells was assessed using MTT assay. All results were shown as mean ± standard deviation. *P < 0.05, **P < 0.01, ***P < 0.001 versus designated group.

Western blotting

Western blot analysis revealed that compared to the control groups, drug treatment increased the protein levels of P53, BAX, and caspase3 (P < 0.05 or 0.01) in endometriotic lesions (Fig. 12).

Figure 12 The results of Western blotting about different targets.

The group 1–3 represented control group, and the group 4–6 represented GZFLW group. All results were shown as mean ± standard deviation. ***P < 0.001 versus designated group.

Apoptosis analysis by flow cytometry

Compared with the control group, the percentage of apoptotic cells in the GZFLW group with the dose of 4 mg increased significantly (Fig. 13).

Figure 13 The apoptosis analysis of flow cytometry.

The analysis of control group and GZFLW group was assessed by AnnexinV/PI staining using flow cytometry. The result was shown as mean ± standard deviation. *P < 0.05 versus designated group.

Discussion

As we all know, TCM is characterized by multiple components, multiple targets, and multiple pathways in the treatment of diseases (Lang et al., 2018). Because of the complex composition, TCM’s clinical and pharmacological research is often difficult to carry out. Network pharmacology is a systematic research method, suitable for the “multi-component, multi-target, multi-pathway” synergistic characteristics of TCM. In this study, we are the first to explore the possible therapeutic mechanism of GZFLW on EMs through network pharmacology and molecular docking methods, to provide directions and ideas for further experimental research.

The major active ingredients of GZFLW

Based on the Hub nodes-Compound Network, we found a few hub compounds: quercetin and kaempferol contained in CM, taxifolin contained in CT, pachymic acid contained in PC, ellagic acid, baicalein, and paeoniflorin contained in RPR, campesterol, hederagenin contained in PK, β-sitosterol, contained in CM, RPR and PK. Among them, quercetin, kaempferol, taxifolin, and baicalein are flavonoids that have shown potent anti-EMs activity by effectively relieving symptoms and inhibiting levels of CA-125 (Signorile, Viceconte & Baldi, 2018). Quercetin and kaempferol can significantly reduce the ailing area of the endometrium via anti-proliferation and anti-inflammatory effects (Park et al., 2019) in EMs mice (Ilhan et al., 2020). Previous studies have demonstrated that phytosterols, including β-sitosterol and campesterol, contributed to the regression of EMs (Ilhan et al., 2019). Baicalin has also been reported to treat EMs by reducing the activity of endometrial stromal cell(ESC) (Jin, Huang & Zhu, 2017), which provided evidence for our results. In conclusion, we speculate that these above ingredients are the potential material basis of GZFLW in treating EMs.

It is mainly indicated that the pathways may be associated with treating EMs of GZFLW were regarding the immune response, apoptosis and proliferation, oxidative stress, and angiogenesis.

Inflammatory response

The differentially expressed genes are related to inflammation in EMs (Ahn et al., 2016), which suggest its role in the course of EMs. Prostaglandin-endoperoxide synthase (PTGS2), also known as COX-2, is related to the pain and infertility of EMs and the PTGS2/ PGE2 axis is considered as the critical target during EMs (Li et al., 2020). TNF, IL1- β are the other hub targets in this study. The immunoreactivity of which induces inflammation in the peritoneal fluid of EMs through activating NF-kappa B and mitogen-activated protein kinase (MAPKs) signaling pathways (Kralickova et al., 2018). To some extents, components such as hederagenin, campesterol, β-sitosterol, pachymic acid, quercetin, and taxifolin in GZFLW inhibit the expression of inflammatory factors to influence the whole inflammatory response. Previous studies have demonstrated that hederagenin inhibits the expression of inflammatory factors, including TNF, IL1, IL6, COX-2 via MAPKs and NF-kappa B pathway (Akhtar et al., 2019; Kim et al., 2017; Lu et al., 2015). Campesterol, β-sitosterol, known as sterols, can reduce the levels of TNF-α and IL of peritoneal fluid in the EMs mice (Ilhan et al., 2019). Flavonoids have a similar effect. Studies have experimentally proved that quercetin has an anti-inflammatory effect on the EMs (Park et al., 2019). Coincidentally, the docking result confirmed all the above, which also provides certain credibility for our results that GZFLW possibly worked against inflammatory in a multi-ingredient way.

Apoptosis and proliferation

The cell apoptosis’s inhibition and concomitant cell excessive proliferation play some important roles in the development of EMs. We think that the expression and regulation of the Jun and Fos genes are among the key factors in cell proliferation and apoptosis. It is found that high expression of c-Jun related to the proliferation of EMs cells via the JNK / c-Jun signaling pathway (Yu et al., 2018). TP53 is another critical tumor suppressor gene in our study; its regulative effect on the cell cycle plays a crucial role in obtaining proliferation activity for ESC of EMs (Hirakawa et al., 2016). It is investigated that the polymorphisms of TP53 may be involved in high risk of the generation of EMs (Hussain et al., 2018). Epidermal growth factor (EGF) also plays a vital role in regulating cell growth, proliferation, and differentiation. Previous studies have found that the EGF expression in patients with severe EMs significantly increased (Chatterjee et al., 2018). Furthermore, there are researches shown that COX-2 expression regulated by the PTGS2 gene can activate MAPKs, which then subsequently activate some transcription factors and protein kinases, ultimately promoting the proliferation of cells.

GZFLW’s regulative effect on cell proliferation and apoptosis in EMs is possibly achieved by regulating flavonoids on the expression of PTGS2 (Da Silva et al., 2020), which further regulates the COX-2/PGE2 axis (Takaoka et al., 2018). Besides, components such as flavonoids, quercetin in GZFLW affect the proliferation and apoptosis in EMs by inhibiting the ERK1/2, P38 MAPK, and AKT signaling pathway (Park et al., 2019). Also, baicalin can affect the activity of ESC by apoptosis inhibition via the NF-kappa B signaling pathway (Jin, Huang & Zhu, 2017). Furthermore, the direct or indirect inhibition of natural triterpenoids on proliferation is possibly another essential component of the regulation of cell apoptosis (Chen et al., 2019).

It is mentioning that all the primary active ingredients except paeoniflorin had a high docking score with EGF in our docking results. Besides, three active ingredients (hederagenin, phytosterol, and pachymic acid) had tight combinations with Jun, FOS, and PTGS2, and the other three active ingredients (pachymic acid, taxifolin, and hederagenin) had strong putative interaction with P53, which demonstrates that GZFLW is likely to affect on proliferation and apoptosis through a multi-ingredient synergistic way.

To verify the effect of GZFLW on EMs, we conducted in vitro experiments. The results of MTT and flow cytometry confirmed that GZFLW can effectively promote the apoptosis of ectopic stromal cells in EMs. In the treatment group, the expression levels of P53, caspase3 and BAX were significantly up-regulated, suggesting that the therapeutic effect of GZFLW may be related to the regulation of P53 pathway and apoptosis pathway.

Oxidative stress

More and more evidence has shown that the pathophysiological mechanism of EMs is complicated, and hypoxia plays an essential role in EMs damage (Wu, Hsiao & Tsai, 2019). The AGE/RAGE and HIF-1alpha signaling pathway were speculated as probable mechanisms of GZFLW against EMs from KEGG results. Previous studies have demonstrated that high expression of RAGE under hypoxia environment increased the accumulation of ROS in peritoneal fluid (Polak et al., 2018). Eventually, contribute to the cell invasion ability in EMs (Seguella et al., 2019). It is also found that high expression of HIF-1α in the serum of EMs aggravated the severity of dysmenorrheal (Zhang et al., 2018).

Fortunately, triterpenoids and flavonoids including hederagenin, β-sitosterol (Adebiyi et al., 2019; Ponnulakshmi et al., 2019), campesterol (Alvarez-Sala et al., 2018; Eom et al., 2017; Ho, Liu & Loke, 2016; Ogunlaja et al., 2016; Shamloo et al., 2017; Toiu et al., 2019; Weingartner et al., 2017), taxifolin, and pachymic acid were reported to be the natural antioxidants, which could reduce oxidative stress by inhibiting RAGE expression to a certain extent (Song et al., 2017) via MAPK, and PI3K/Akt pathways (Wen et al., 2018). The molecular docking results showed that triterpenoids and flavonoids have tight combinations with the hub targets AKT1, MAPK1, and MAPK8. However, further experiments are needed to confirm whether GZFLW worked against EMs by affecting AGE/RAGE and HIF-1α via MAPK and PI3K/Akt pathways.

Angiogenesis

Hypoxia and inflammatory, as an essential characteristic of EMs microenvironment, can stimulate the transcription of hypoxia-inducible factor HIF (Lin et al., 2018) and inflammatory-inducible factor COX-2 gene, consequently activate the angiogenesis through the PI3K/mTOR and MAPK pathways. HIF-1α/ VEGF expression in serum and endometrium has been reported to have a relationship with the stage of EMs and the severity of dysmenorrheal (Zhang et al., 2018). Studies confirmed that some plant inhibitors of COX-2 were able to treat EMs and inhibited ESC’s angiogenesis by regulating HIF-1α/VEGF in mice models (Li et al., 2020).

It is worth mentioning that triterpenoids (β-sitosterol, campesterol, hederagenin) and flavonoids (taxifolin) in GZFLW were detected in our study, they could have a modest effect on inhibiting VEGF in EMs, which have been confirmed by experiments and in line with our molecular docking results (Ilhan et al., 2019). Previous studies have also demonstrated that paeoniflorin could inhibit angiogenesis via the HIF-1 α/ VEGF pathway (Song et al., 2017; Zhou, Ding & Hardiman, 2018).

Conclusions

In this study, network pharmacology and molecular docking were utilized for exploring the potential effects of GZFLW in anti-EMs, mainly focusing on four aspects: the suppression of inflammatory response, the regulating of apoptosis and proliferation, the reduction of the oxidative stress, and the inhibition of angiogenesis. Among them, the pro-apoptotic effect was confirmed by our in vitro experiments. In these crucial biological functions, triterpenoids and flavonoids contained in the GZFLW were considered that those might be the critical active ingredients involved in the potential therapy mechanisms.

Our study had several limitations. Firstly, to some extent, network pharmacologic analysis can help to predict the possible molecular mechanism of GZFLW in treating EMs. However, we can only speculate but not confirm if such mechanisms have an impact because, in evidence-based medicine, the conclusion can not be determined until subsequent biological experiments. Secondly, the data set of the GZFLW related compounds and targets were identified through the TCMSP, BATMAN-TCM, and CASC database, and GeneCards built the EMs target database. Choosing different databases may lead to the study biases. Moreover, because all the databases we use are based on existing research results, this may limit the discovery of new targets related to EMs treatment.

In conclusion, our study reveals that GZFLW may have a multiple effect on EMs. Although the mechanism remains confirmed by further experiments, the relevant targets and signaling pathways for GZFLW against EMs have been preliminarily studied and systematically summarized.

Supplemental Information

Supplemental Information 1 Raw data

Click here for additional data file.

Additional Information and Declarations

Competing Interests

Author Contributions

Data Availability

The authors declare there are no competing interests.

Haoxian Wang and Gang Zhou performed the experiments, prepared figures and/or tables, and approved the final draft.

Gang Zhou and Xianyun Fu conceived and designed the experiments, authored or reviewed drafts of the paper, and approved the final draft.

Wei Wang conceived and designed the experiments, analyzed the data, prepared figures and/or tables, and approved the final draft.

The following information was supplied regarding data availability:

Raw data are available in the Supplemental Files.

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
