# Peer review of "Utilizing network pharmacology and molecular docking to explore the underlying mechanism of Guizhi Fuling Wan in treating endometriosis"

_PeerJ, doi:10.7717/peerj.11087_

## Round 0.1 · original submission · Major Revisions

Both reviewers think your work is interesting and meanwhile propose some suggestions to improve. Please address the reviewer's concern point by point.

·

Basic reporting

Wang and Zhou et al conducted a nice in silico network biological study to elucidate the potential mode of action of Guizhi Fuling Wan in endometriosis.

The English of the manuscript is professional, but a little polishing may be needed.

The background and the importance of the topic in the introduction are established.

Please add the database citations, not just the links (Line 104, 105, 114).

Experimental design

I have a few major problems with the manuscript.

1. Both the BATMAN-TCM database and the TCMSP database are using predicted targets.
The manuscript considers these targets as the "real" targets of the TCM compounds. We cannot be sure about it. The authors must consider this. This is a major limitation of the manuscript.

2. The authors used a network biological technique to find the hub proteins between their targets. It resulted already well-known hubs e.g. p53, EGFR, EGF, AKT1. The authors miss mentioning the study biases and network analysis in the discussion.


3. The network figures need to be redrawn. I strongly suggest for the authors to read Marai et al visualisation guide about network figures. https://journals.plos.org/ploscompbiol/article?id=10.1371/journal.pcbi.1007244
In details:
Figure 2: The mol numbers do not contain information for the reader. Please use the name of the molecules.
Figure 4 and 5: Both networks do not contain information. I am against to add a network figure only for degree selection. Please use a force-directed layout or spring embedded layout for it. That should draw the pathways and molecular functions which are involved in the putative mode of action of GZFLW. The large betweenness and degree nodes can be highlighted in the network. Does the network have multiple components? Are the ingredients of GZFLW are acting on the same part of the network? The network can be coloured by the five ingredients. This way it can show which way each of the 5 ingredients acts. Those targets that are targeted by more than 1 ingredient can be coloured by an additional colour.
Figure 6 The authors tried to answer which compound act on which hub protein. The double circle layout does not help. I suggest instead two columns/rows the highest degree compound/protein at the top. This way the interactions will be a bit more clear. Also, the compound names need to be on the figure just like in Figure 2.
Figure 7 I am not satisfied with the layout. A non-circular layout would be better. It would show the interactions between pathways and targets better. The figure by itself does not contain extra information a table can be as good. But I have another underlying problem of this part.

4. For the GO and the KEGG enrichment analysis (DAVID/METASCAPE) the backgrounds are not mentioned. The problem with this is the following: Tha authors filtered proteins which are involved in endometriosis and which are drug targets. The first is done by the filtering to the disease. The second is based on the used databases. Any drug-target database will only have the druggable genome. The intersection of these two sets will be highly overlapping to the specific targets of GZFLW.
The problem in the sense of pathway enrichment analysis we will hardly get any specific result to GZFLW, it is possible it is coming just by the filtering criteria. (endometriosis, drugable genome) To see which pathways/molecular functions are specific for the compound combination I suggest using an enrichment analysis of all the endometriosis targets from BATMAN-TCM database and the TCMSP.

5. If possible I suggest having more thorough docking study for each target.

6. Please describe where are the oral bioactivity and drug-likeness are coming from, and explain why these cut-offs were chosen.

Validity of the findings

I have not much to add except winch I described on the experimental design. The limitations of the study need to be addressed.

The compounds and their putative targets are discussed in the discussion. However, the targets are putative and this needs to be emphasised.

Additional comments

Please add a space before the citations.

Reviewer 2 ·

Basic reporting

1. The introduction needs more details. It is suggested that the authors improve the description at lines 71- 81 to make the introduction of Traditional Chinese medicine (TCM) network pharmacology clear and have literature well referenced.
2. In addition, to improve the readability, the language of this manuscript should be improved, e.g., the meaning of "multi-component target medicine" is ambiguous in Page 3.

Experimental design

There are some unclear points described in Method section.
1. The collection of GZFLW ingredients should consider some other high-quality databases, then the results from different databases should be sufficiently compared to obtain more complete and reliable GZFLW ingredients.
2. The GZFLW targets prediction in this manuscript is rather like targets collection. How to ensure the reliability of GZFLW targets for further study should be sufficiently considered.
3. In addition, the definition of “Betweenness Centrality” (Line 120) should be checked again.
4. How to condiser the traditional efficacy of GZFLW in treating endometriosis? It is suggested that authors take into consideration of the traditional efficacy of GZFLW based on the biological network related to TCM syndrome (ZHENG), for example, the Cold Syndrome network or the Hot Syndrome network (IET Syst Biol 2007 [PMID: 17370429]).

Validity of the findings

1. In discussion section, the expression of “immune response” and “Inflammatory response” should be unified.
2. It is not enough to just list the analysis results of GO and Pathway enrichment and infer that combined synergetic effects of these pathways were the mechanisms of GZFLW. It should be provided with several aspects of solid evidence to support these results.

Additional comments

I suggest authors make major revisions of this paper.

---

## Round 0.2 · Major Revisions

The two reviewers read the updated manuscript again. Both reviewers are not satisfied with the revision. Please take time to carefully revise your paper in the second round. If experimental or clinical validation is difficult, the authors should try to provide some literature or database support and necessary discussion for limitations and future work.

·

Basic reporting

The manuscript improved. I am really happy that the authors have addressed the network visualisation comments of mine. The network figures look much better. However, I still have some points which need to be addressed, mostly English.


I will list here some English mistakes which need to be addressed:
Line 55 extra space
Line 56 extra space
Line 70-71 use instead of "to network target mediated by a multi-component system" use "to multiple interacting targets mediated by multiple compounds."
Line 78 I have never heard the word "systematicness"
Line 84 through the STRING database
Line 118 The active compounds' targets in GZFLW were obtained...
Line 127 The interactions of the targets were obtained, not the targets.
Line 131 Betweenness centrality is the number of the shortest path going through a given node/edge. Sometimes it is measured by the absolute value or the percentage of all shortest paths. - The authors use percentage-based betweenness centrality in their manuscript.
Line 156 KEGG citation missing
Line 157 Cytoscape citation missing
Line 196 performed and resulted in 116 GO...
Line 249 I guess: The differentially expressed genes are related to inflammation in EM.?
Line 287-290 The authors are using "tight combination" instead of high docking score or strong putative interaction. Please use one of the tight combinations is meaningless in this case.

Figure 2 Please add the figure description of the ingredients name. It makes it much easier to read.

Table 2 The Betweenness centrality is not in the table
Table 3 What is the measurement of the docking score? Usually, it is some kind of energy best of my knowledge?

Experimental design

What was the cut-off for "tight combination"?

Validity of the findings

I have here two large issues.

The authors forget to mention after I have asked them in my previous revision that most of the targets of the compounds are predicted, putative targets. In their response, they mention that they wrote in the limitation paragraph, but it is not there.

Another validity question regarding the study is the interaction of the compounds and various ligands such as EGF, VEGEFA, IL-6. It is hard for me to imagine biologically relevant inhibition or effect if the compound directly targets the ligands and not the receptors. The authors specifically mention EGF as their most strongly interacting protein with the TCM ingredients target. Why do they think that the interaction is relevant? What do the TCM compounds do with EGF? Use less strong language, if no further validity is available.

Additional comments

The manuscript improved a lot. Please mention that the databases are mostly predicted interactions! Even after the docking study, the results are suggesting a putative mechanism of action of the TCM combination. It is a nice network biological work but this is the maximum what we can say based on the results.

Reviewer 2 ·

Basic reporting

In the revision, authors did not provide any solid and new evidence from experimental and clinical studies to ensure the reliability of GZFLW targets, GO and Pathways, as well as synergetic effects to support their claims.

Experimental design

no comment.

Validity of the findings

no comment.

Additional comments

no comment.

---

## Round 0.3 · accepted · Accept

The authors followed the minor revision suggestions, the presentation has been further improved.